# Selection Induced Collider Bias in LLMs: A Gender Pronoun Uncertainty Case Study

## Abstract

In this paper, we cast the problem of task underspecification in causal terms, and develop a method for empirical measurement of spurious associations between gender and gender-neutral entities for unmodified LLMs, detecting previously unreported spurious correlations. We then describe a lightweight method to exploit the resulting spurious associations for prediction task uncertainty classification, achieving over $90\%$ accuracy on a Winogender Schemas challenge set. Finally, we generalize our approach to address a wider range of prediction tasks and provide open-source demos for each method described here.

## 1 Introduction and Related Work

This paper investigates models trained to estimate the conditional distribution: $P(Y|X, S)$, where $S$ is the cause of sample selection bias in the training dataset. Selection bias is not an uncommon problem, as most datasets are subsampled representations of a larger population, yet few are sampled with randomization (Heckman, 1979).

### 1.1 Causal DAGs and Biases

Sample selection bias occurs when some mechanism, observed or not, causes preferential inclusion of samples into the dataset (Bareinboim and Pearl, 2012). Employing the language of causal inference, selection bias is distinct from both confounder and collider bias. Confounder bias can occur when two variables have a common cause, whereas collider bias can occur when two variables have a common effect. Correcting for confounding bias requires that one condition upon the common cause variable; conversely correcting for collider bias requires that one does not condition upon the common effect (Pearl, 2009).

The type of selection bias that interests us here is that which involves more than one variable (observed or not), whose common effect results in selection bias. Such assumed relationships can be compactly and transparently represented as a causal data-generating process (DGP) in the form of a directed acyclic graph (DAG), for example illustrated in Figure 1. The absence of arrows connecting nodes in causal DAGs encodes assumptions, for example that $W$ and $G$ in Figure 1(a) are stochastically independent of one another. The direction of the arrowhead encodes our assumptions about the direction of causation. For example, the two arrows departing from $W$ and $G$ toward $S$ encode the assumption that $S$ is a common effect of $W$ and $G$.

In Figure 1, the twice-encircled node, $S$, symbolizes some mechanism that can cause samples to be selected into the dataset. To capture the statistical process of sampling for dataset formation, one must condition on $S$, thus inducing the collider bias relationship between $W$ and $G$ into the DGP.

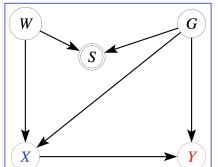 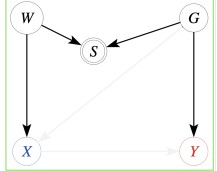

(a) Well-specified: Symbolic entity $G$, a common cause of both dataset features and labels.

(b) Underspecified: $G$ unobserved by features, thus features contain no causes of the labels.

Figure 1: Data generating process for high dimensional data, such as in NLP, where $X$ and $Y$ represent high dimensional text features: the dataset features and labels, while $W$, $G$, and $S$ represent low dimensional symbolic entities that may cause the text.

We will use the term *selection induced collider bias* to refer to circumstances such as this one, when the selection bias mechanism induces a collider bias relationship in the dataset that would not have been there otherwise[1]. Selection induced collider bias has been covered in medical and epidemiological literature (Griffith et al., 2020) (Munafò et al., 2018) (Cole et al., 2009) and received extensive theoretical treatment from Pearl and Bareinboim in (Bareinboim and Pearl, 2012), (Bareinboim et al., 2014), (Bareinboim and Tian, 2015) and (Bareinboim and Pearl, 2016), yet has received very little attention in deep learning literature.

## 1.2 Underspecification and Spurious Associations

We define a learning task as *underspecified* when none of the features available to the model (at training or inference time) are causes of the label. Figure 1(b) encodes this relationship with the absence of an arrow between features, $X$, and labels, $Y$. With no causal features available, models must resort to learning any spurious associations that will reduce predictive risk, regardless of how tenuous the association may be. We refer to these as *otherwise non-interacting* spurious associations.

We would like to draw a distinction between the type of spurious association induced by underspecification, and the spurious associations most often addressed in today's literature. For example, the task of predicting cow vs camel (perhaps based on *spurious* grassy vs sandy background pixel features), would not be considered an underspecified task, due to the availability of the *causal* cow vs camel pixel features in the foreground. From a causal perspective, the symbolic background entity is a common cause of both the pixel features and the labels, inducing confounder bias and thus the learning of spurious associations along a *secondary* path (Arjovsky et al., 2019), in addition to the *primary* direct causal path from feature to label.

A natural question to ask is, how does spurious association flow from $X$ to $Y$, if not through some confounding variable like background, nor though a direct causal path. As demonstrated in (D'Amour et al., 2020), weakly-interacting prediction tasks display significant variance, even due to changes in the random seed initialization. In this work, by focusing on variables engaged in a relationship of selection induced collider bias, we are able to open up a *tertiary* path between $X$ and $Y$: the path along $X \leftarrow W \rightarrow S \leftarrow G \rightarrow Y$ in Figure 1(b). In distinction to (D'Amour et al., 2020), we argue this causal perspective facilitates the identification of otherwise non-interacting (and previously unreported) spurious associations, and importantly enables the injection of these 'benign' spurious tokens into text at inference time, to achieve an uncertainty measurement.

## 2 Contributions

In this paper we make the following contributions:

- We cast the problem of task underspecification in causal terms and apply causal inference methods to hypothesize the effects of selection induced collider bias on underspecified tasks.
- We test these hypotheses on unmodified and widely used pre-trained LLMs via a case study of gender pronoun resolution, resulting in two new findings:
    - A method for empirical measurement of spurious correlations between gender and gender-neutral entities for unmodified LLMs which permits measurement of previously unreported spurious correlations between gender vs location and time.
    - A method for quantifying inference-time task uncertainty with an accuracy of over $90\%$ when testing RoBERTa-large with the Winogender Schema challenge test set.
- To demonstrate that both above methods are reproducible, lightweight (dozens of lines of code), time-efficient (takes seconds), and plug-n-play compatible with almost any BERT-like LLM, we provide open-source and running demos:
    - Spurious Correlations: https://huggingface.co/spaces/paper5186/spurious.
    - Uncertainty: https://huggingface.co/spaces/paper5186/uncertainty.
- We generalize our approach to address a wider range of prediction tasks and provide results on a generic DGP that are consistent with our empirically measured results on LLMs.

---

[1]Although conflated, collider bias can occur independent of selection bias and vice versa (Hernán, 2017).

## 3  PROBLEM SETTING: GENDER PRONOUN RESOLUTION TASKS

An example of underspecification can be found in the gender pronoun resolution task in Figure 2(b)[2] Gender pronoun resolution will serve as a case study for the rest of this paper, for largely two reasons: 1) it is a well-studied problem with many recent advances (Cao and Daumé III, 2020), (Lu et al., 2018), (Webster et al., 2020), (Zhao et al., 2018) and yet remains a challenging problem for modern LLMs (Deng et al., 2022) and 2) it provides many human-relatable underspecified task scenarios.

To apply causal inference methods to gender pronoun resolution, we must make some initial assumptions about a plausible DGP for the relevant features and labels, that we can later support with empirical measurement. These assumptions

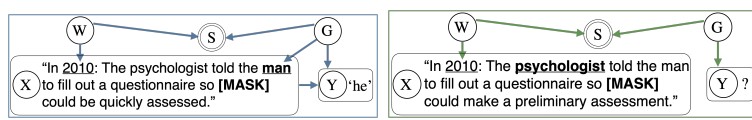

(a) Well-specified ($X \rightarrow Y$): Masked pronoun coreferent with the *man*.

(b) Underspecified ($X \nrightarrow Y$): Masked pronoun coreferent with *psychologist*.

Figure 2: Gender pronoun resolution task with modified Winogender Schema (Rudinger et al., 2018) test sentences.

have been transparently represented in Figure 1 and Figure 2. Note the heterogenous nature of the DAG variables, in which $X$ and $Y$ represent high dimensional features like text in the dataset, while $W$, $G$, and $S$ represent low dimensional symbolic entities that may cause the text.

Specifically in Figure 2, the symbol $X$ represents the *text* sentences in our dataset, and $Y$ represents the label: a *gender pronoun*. The arrow pointing from $X$ to $Y$ encodes our assumption that $X$ is more likely to cause $Y$, rather than vice versa. The remaining symbols in Figure 2 are not recorded in the dataset. The symbol $G$ represents *gender* and in well-specified gender resolution tasks, $G$ causes both $X$ and $Y$. $W$ represents gender-neutral entities that are not the cause of $Y$, but still of interest because they cause $X$. Additionally, in order to measure the effects of selection induced collider bias, we must find entities for $W$ that are also the cause of $S$: selection into the dataset.

The $W \rightarrow S \leftarrow G$ relationship can represent any selection bias mechanism that has a gender dependency upon otherwise gender-neutral entities. For example, in data sources like Wikipedia written about people, it is plausible that *access* (e.g. access to resources) has become increasingly less gender dependent as the *date* approaches more modern times, but not evenly in every *place*. In data sources like Reddit written by people, $W \rightarrow S \leftarrow G$ can plausibly capture the scenario that even in the case of subreddits about *gender-neutral hobbies*, the style of the moderation and community may result in gender-disparate *access* to a given subreddit. In both cases, the disparity in *access* can result in preferential inclusion of samples into the dataset, on the biases of gender. In this paper, we use $W$ to represent *date* and *place*, for our LLM measurements.

Figure 1(b) and Figure 2(b) are the underspecified counterparts to the well-specified prediction tasks in Figure 1(a) and Figure 2(a). As defined above, to achieve the underspecification of interest here, we must obscure any causal features from $X$. In the case of gender pronoun resolution, this is achieved by removing the path between $G$ and $X$. Further, because $W$ is also gender-neutral, once we have removed any gender-identifying features from $X$, we should additionally remove the path between $X$ and $Y$, as there is no longer any feature in $X$ causing $Y$.

With the causal path from $X \rightarrow Y$ removed, we would then expect $X$ and $Y$ to be unconditionally independent in the real world ($RW$). However, in the scenario depicted in Figure 1(b) and Figure 2(b), selection induced collider bias has opened the backdoor path, $X \leftarrow W \rightarrow S \leftarrow G \rightarrow Y$. Thus, the learned model ($LM$) is encouraged to learn any spurious association along this path that reduces predictive risk, resulting in $X$ and $Y$ becoming unconditionally dependent.

Formally, selection induced collider bias causes the transformation: $(Y \perp\!\!\!\perp X)_{RW} \overset{s}{\Rightarrow} (Y \not\perp\!\!\!\perp X)_{LM}$, due to the spurious path through $S$: $X \leftarrow W \rightarrow S \leftarrow G \rightarrow Y$. Whereas spurious associations learned in the well-specified model would more likely travel a path directly through a confounding shortcut feature, such as a sandy or grassy background, as was discussed previously in the camel vs cow example.

---

[2]A single prediction task may be partitioned into well-specified and underspecified 'sub'tasks. For example, Figure 2(b) may be well-specified for part-of-speech tagging.

## 4   Method: Spuriousness due to Selection Induced Collider Bias

In addition to transparently stating our assumptions, the causal DAGs in Figure 1 and Figure 2 entail specific conditional probabilities that should be empirically measurable, under the condition of two additional assumptions: For a given causal graph, $Gr$, and a measured probability distribution $Pr$, we are assuming 1) the Markov assumption: $(X \perp\!\!\!\perp Y | Z)_{Gr} \Longrightarrow (X \perp\!\!\!\perp Y | Z)_{Pr}$ and 2) faithfulness: $(X \perp\!\!\!\perp Y | Z)_{Gr} \Longleftarrow (X \perp\!\!\!\perp Y | Z)_{Pr}$ (Peters et al., 2017), where $Z$ is the set $\{W, G, S\}$.

To capture the statistical process of dataset formation, we implicitly condition on $S = 1$ for all the samples in the dataset. Conditioning on $S$ induces collider bias between $G$ and $W$ in the form of $S$'s structural equation (Pearl, 2009): $S := f_s(W, G, U_s)$ (where $U_s$ is the exogenous noise of the $S$ variable), entailing a conditional dependency between otherwise non-interacting entities: $G, W$.

However, at inference-time we only have access to $X, Y$. Here we will show that for underspecified tasks, we would expect $P(Y|X)$ to be distributed similarly to $P(G|W)$. Revisiting the underspecified DAGs in Figure 1(b) and Figure 2(b), and applying the Markov and faithfulness assumptions, we can estimate the conditional probability of a gender pronoun, $Y$, given gender-neutral text, $X$.

Equation (1) shows a mapping from the target unbiased quantity to the measured selection biased data, as defined in (Bareinboim and Pearl, 2012). Equation (2) assumes very high correlation between the textual form of gender in $Y$ (as a *gender pronoun*), with the symbolic variable for gender, $G$. Equation (3) replaces $S$ with the variables in its structural equation, $S := f_s(W, G, U_s)$, which entails the conditional dependence

$$
\begin{aligned}
P(Y|X) &= P(Y|X, S{=}1) &\quad (1)\\
&\sim P(G|X, S{=}1) &\quad (2)\\
&\sim P(G|X, W) &\quad (3)\\
&\sim P(G|W) &\quad (4)
\end{aligned}
$$

$P(G|W) \neq P(G)$, and thus we add $W$ behind $G$'s conditioning bar. Finally, Equation (4) assumes we have an underspecified (gender-neutral) text, $X$, so $P(G|X) = P(G)$, and thus we remove $X$ from behind the conditioning bar.

Equation (1) - Equation (4) show that $P(Y|X) \sim P(G|W)$ in underspecified tasks, providing information about the otherwise inaccessible latent representations for $G$ and $W$, and specifying a measurable relationship between $X$ and $Y$, that we can validate in the next section.

### 4.1   Experimental Setup: Masked Gender Task (MGT) Challenge Set

To validate spurious associations between *otherwise non-interacting* entities, such as *time* and *gender*, we desire a challenge test set of gender-neutral text. However, the text from the example in Figure 2(b) does not satisfy this requirement due to associations between gender and occupation. We also desire a test set compatible with the Masked Language Modeling (MLM) objective used in the pretraining of LLMs[3], for the greatest applicability of the results. Table 1 shows the heuristic used to create the MGT challenge set composed of $(2 \text{ Python f-string templates}) \times (5 \text{ tenses of } \texttt{verb} \text{ to-be}) \times (6 \texttt{ life\_stages}) \times (30 \text{ values for } W \text{ as } date) \times (20 \text{ values for } W \text{ as } place) = 36000$ gender-neutral test sentences.

In Section 4.2, each plotted dot is the softmax probability (averaged over 60 gender-neutral texts) for predicted gender pronouns. These are plotted against *year* (or *country*), where the *year* (or *country*) along the x-axis matches the gender-neutral $W$ value injected into the gender-neutral text[4].

As the final layer in the pre-trained LLM is a softmax over the entire tokenizer's vocabulary, we sum (without normalization) the gender-identified portion (as listed in Table 3) of the probability mass from the top five predicted words[5].

### 4.2   Results: Pre-trained LLMs Gender-Date and Gender-Place Dependencies

Figure 3 shows pre-trained BERT (Devlin et al., 2018) and RoBERTa (Liu et al., 2019), base and large, results following the experimental setup described above.

---

[3]Tests compatible with the MLM objective can avoid fine-tuning weights during measurement procedures.

[4]For example, the purple and green dots at the x-axis position of 1938 are the female and male pronoun softmax probabilities for the masked word in input texts like 'In 1938, [MASK] will became a teenager.'.

[5]We pick the number $k = 5$ for the 'top_k' predicted words, because 5 is the default value for the 'top_k' argument in the Hugging Face 'fillmask()' function used for inference.

Table 1: Heuristic for creating gender-neutral input texts for the MGT challenge set, and example rendered text. For `verb` we used past, present and future tenses of the verb *to be*: ['was', 'became', 'is', 'will be', 'becomes'], and for `life_stage` we used proper and colloquial terms for a range of life stages: ['a child', 'a kid', 'an adolescent', 'a teenager', 'an adult', 'all grown up'], excluding life stages past adulthood due to age gender-inequities. For the text versions of `w`, we used a range of 30 *dates* and 20 *places*, defined in Appendix C.

| $W$ Category | Python f-string templates | Example text |
|---|---|---|
| Date & Place | `` `f"[MASK] {verb} {life_stage} in {w}."' `` 
 `` `f"In {w}, [MASK] {verb} {life_stage}."' `` | '[MASK] was a teenager, in 1953.' 
 'In Mali, [MASK] will be an adult.' |

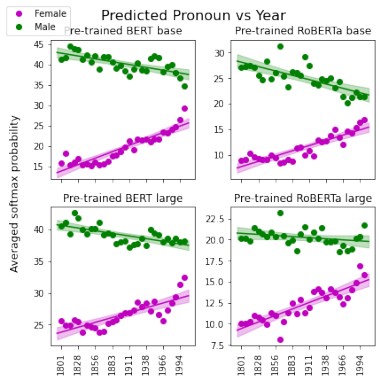
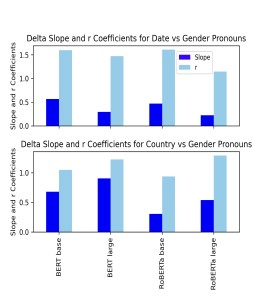
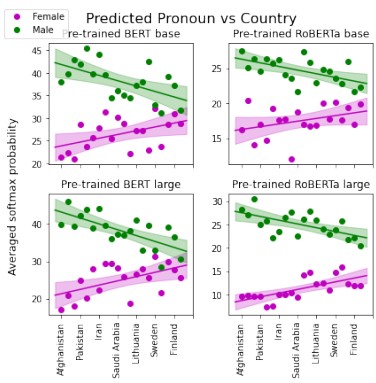

(a) Spurious correlation between *gender* and *time* plotted as averaged softmax probabilities for predicted gender pronouns vs a range of dates.

(b) Difference, between the female and male pronoun, linear fit slope and Pearson's $r$ coefficients for Figure 3(a) (top) and Figure 3(c) (bottom).

(c) Spurious correlation between *gender* and *place* as list of countries, ordered by their Global Gender Gap rank (see Appendix C.1).

Figure 3: Softmax probabilities and linear fit coefficients for LLM predictions displaying spurious correlation between *gender*, $G$, and gender-neutral entities, $W$, injected into gender-neutral input texts described in Table 1.

The shaded regions in Figure 3(a) and Figure 3(c) show the 95% confidence interval for the linear fit, and Figure 3(b) shows the slope and Pearson's $r$ correlation coefficient (following (Rudinger et al., 2018)) of the y-axis value against the *index* of the x-axis, for these plots. It is noteworthy that the magnitude of the spurious correlations appears similar across model size, but from this data alone, it is unclear the role of model overparameterization in the learning of spurious associations due selection induced collider bias.

Overall, in Figure 3 we can see that the injection of different $W$ values into $X$ results in a changed prediction for $Y$ consistent with the $P(G|W)$ conditional dependency entailed by collider bias at $S$. These spurious associations of *gender* vs *time* and *place* have not been reported before this work. Further, with the derivation that $P(Y|X) \sim P(G|W)$ in underspecified tasks, additional such spurious associations can be more easily identifiable and measurable using the inference-time method introduced here.

## 5    METHOD: EXPLOITING FOR UNCERTAINTY MEASUREMENT

In (Bareinboim et al., 2014) it is proven that one can recover the unbiased conditional distribution $P(Y|X)$ from a causal DAG, $G_S$, with selection bias: $P(Y|X, S = 1)$, if and only if the selection mechanism can become conditionally independent of the effect, given the cause: $(Y \per\!\!\!\perp S|X)_{G_S}$. However, in Figure 1(b) we can see $(Y \not\perp\!\!\!\perp S|X)$ trivially, due to the lack of a direct path between $X$ and $Y$. Thus, the conditional distribution learned on an underspecified task with selection induced collider bias will not converge toward the unbiased distribution without additional data or assumptions (Bareinboim et al., 2014). However, in this section we will exploit the outcomes of selection bias to our benefit, in a method we call $W$-injection and describe below.

Table 2: Extended Winogender evaluation sentences and uncertainty results for occupation as 'Doctor'. Only sentence IDs 5 & 6 are well-specified for gender pronoun prediction, and appropriately all models show the lowest uncertainty for these sentences. For example, a cutoff threshold of 1.0 for our uncertainty metric would correctly classify each sentence as well or underspecified.

| | | Uncertainty | | | |
|---|---|---|---|---|---|
| | | BERT | | RoBERTa | |
| ID | Sentence with Participant and **Coreferent** Highlighted | base | large | base | large |
| 1 | In DATE: The **doctor** told the man that [MASK] would be on vacation next week. | 1.7 | 1.8 | 15.0 | 14.0 |
| 2 | In DATE: The **doctor** told the woman that [MASK] would be on vacation next week. | 4.3 | 27.3 | 4.0 | 18.8 |
| 3 | In DATE: The **doctor** told someone that [MASK] would be on vacation next week. | 10.6 | 8.0 | 13.3 | 20.2 |
| 4 | In DATE: The **doctor** told the patient that [MASK] would be on vacation next week. | 1.9 | 6.6 | 14.7 | 16.6 |
| 5 | In DATE: The doctor told the **man** that [MASK] would be at risk without the vaccination. | **0.0** | **0.0** | **0.3** | **0.1** |
| 6 | In DATE: The doctor told the **woman** that [MASK] would be at risk without the vaccination. | **0.3** | **0.1** | **0.7** | **0.5** |
| 7 | In DATE: The doctor told **someone** that [MASK] would be at risk without the vaccination. | 11.3 | 10.5 | 41.3 | 16.4 |
| 8 | In DATE: The doctor told the **patient** that [MASK] would be at risk without the vaccination. | 6.1 | 12.3 | 19.2 | 9.3 |

## 5.1 EXPERIMENTAL SETUP: EXTENDING WINOGENDER TEXTS AND USING SPURIOUSNESS

We test $W$-injection using the Winogender Schema evaluation set (Rudinger et al., 2018), composed of 120 sentence templates, hand-written in the style of the Winograd Schemas (Levesque et al., 2012), and already seen in Figure 2.

The 'Sentence' column in Table 2 shows example texts from our extended version of the Winogender evaluation set, where the occupation is 'doctor'. Each sentence in the evaluation set contains: 1) a *professional*, referred to by their profession, such as 'doctor' 2) a context appropriate *participant*, referred by one of: {'man', 'woman', 'someone', *other*} where *other* is replaced by a context specific term like 'patient', and 3) a single pronoun that is either coreferent with (1) the *professional* or (2) the *participant* in the sentence (Rudinger et al., 2018). As was the case in the MGT challenge set, this pronoun is replaced with a [MASK] for prediction.

Our extensions to the evaluation set are two-fold: 1) we add {'man', 'woman'} to the list of words used to describe the *participant* in order to add well-specified tasks to the existing Winogender set, which are all underspecified, and 2) we perform $W$-injection by prepending each sentence with the phrase 'In DATE'[6], where 'DATE' is replaced by a range of years from 1901 to 2016[7], similar to what was done in Figure 3(a).

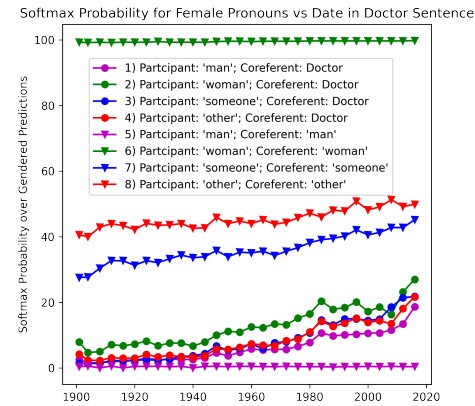

Figure 4: Averaged softmax percentages from RoBERTa large for predicted female gender pronouns (normalized over all gendered predictions) vs a range of dates (injected into the text), for the 'Doctor' Winogender texts listed in Table 2.

In Sentence IDs 1 - 4 of Table 2, the masked pronoun is coreferent with the *professional*, who is always referred to as the 'doctor'. Whereas in Sentence IDs 5 - 8, the masked pronoun is coreferent with the *participant*, who is referred to as {'man', 'woman', 'someone', and 'patient'}, respectively. Thus, of the eight sentences, six are underspecified for the pronoun prediction task, with only IDs 5 & 6 as well-specified. An uncertainty metric should only show low uncertainty for IDs 5 & 6.

---

[6]Similar results can be obtained with the $W$-injection of countries, as was done in Figure 3(c).

[7]We picked a slightly narrower and more modern date range as compared to that of Figure 3(a) for semantic consistency with some of the more modern occupations in the Winogender evaluation set.

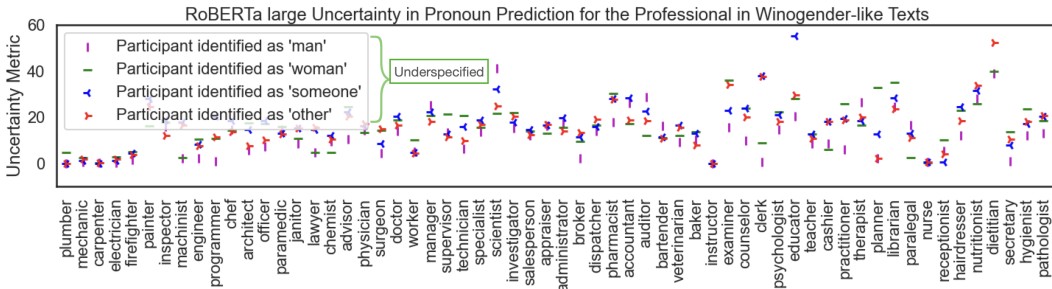

(a) Masked pronoun is coreferent with the *professional* (similar to IDs 1-4 in Table 2), so all these sentences are underspecified.

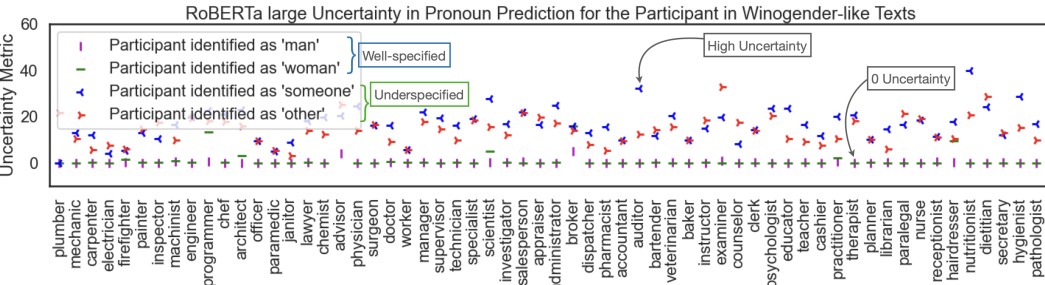

(b) Masked pronoun is coreferent with the *participant* (similar to IDs 5-8 in Table 2), thus sentences containing 'man' and 'woman' are well-specified, and the rest are underspecified.

Figure 5: RoBERTa-large uncertainty metric on all Winogender Schema occupations. Setting a threshold at 1.0 for the metric, produces greater than $90\%$ outcomes for both the true-positive classification of the underspecified tasks and true-negative classification of well-specified tasks.

## 5.2 Results and Design of a Simple Uncertainty Metric

Figure 4 shows the predicted softmax probability for female pronouns for the masked words in the Table 2 sentences, normalized to the gendered predictions of the top five predicted words from pre-trained RoBERTa large. Similar to the findings in (Rudinger et al., 2018), at any given $W$ value, the softmax probabilities for female pronouns are higher for masked pronouns coreferent with the patient as opposed to the doctor, indicating the now well-known learned spurious association between gender vs occupation.

Our contribution here is that we see *no* gender-time spurious associations in the well-specified sentence IDs 5 & 6. Further, in Figure 4 we can see that the spurious associations due to the $W$-injection of an unrelated spurious association (time vs gender) appears *additive* with the existing spurious association between occupation and gender.

As a simple and naive single-value uncertainty metric, we can measure the absolute difference between the averaged softmax probabilities for the first and last several dates along the x-axis in Figure 4. For this uncertainty metric, we would expect larger values for underspecified prediction tasks, in which $W$-injection has a larger influence on the prediction. For the predictions in Figure 4, this metric is shown in the 'Uncertainty' columns in Table 2 for all four LLMs studied in this paper. Here we see uncertainty values closest to 0 for well-specified sentence IDs 5 & 6, with a cutoff threshold of 1.0 producing $100\%$ outcomes for both the true-positive classification (the model should be uncertain) of the underspecified tasks, and true-negative classification (the model should not be uncertain) of well-specified tasks, for the sentences in Table 2.

Our extended version of the Winogender Schema contains (60 *professional* occupations ) $\times$ (4 *participant* texts ) $\times$ (30 values for DATE ) $\times$ (2 sentence templates[8]). This totals to 14,400 test sentences, which we provide as input text to the 4 pre-trained models thus far investigated in this paper: BERT base and large, and RoBERTa base and large.

---

[8]One template with the masked pronoun coreferent with the *professional* and the other with the *participant*.

We calculate the above-described uncertainty metric for all 60 occupations in the Winogender evaluation set and show the results from RoBERTa large in 1) Figure 5(a), with input sentences like IDs 1 - 4 where the masked pronoun is coreferent with the *professional*, and 2) Figure 5(b), with input sentences like IDs 5 - 8 where the masked pronoun is coreferent with the *participant*. In these plots the x-axis is ordered from lower to higher female representation, according to Bureau of Labor Statistics 2015/16 statistics provided by (Rudinger et al., 2018), and the y-axis is the uncertainty metric defined in the proceeding paragraphs.

Similar to the 'doctor' example, in Figure 5, we again see high uncertainty for all six of the underspecified tasks, and low uncertainty for the two well-specified tasks, for almost all Winogender occupations. For each occupation, we should and largely do see high uncertainty for all four participants (including sentences containing 'man' and 'woman') in Figure 5(a), and only high uncertainty for two participants (excluding sentences contain 'man' and 'woman') in Figure 5(b).

Concretely, even if we focus exclusively on the most challenging Winogender sentence-types, in which gender-identifying text: 'man' or 'woman' is co-occurring but not coreferent with the masked-out pronoun, with a thresholding value of 1.0 we find the uncertainty metric has a true-positive classification rate of 90.0% for the underspecified sentences, and a true-negative classification rate of 91.7% for the well-specified sentences.

Overall, we see this simple metric can provide accurate task uncertainty classification for unmodified pre-trained LLMs, with only several additional inference runs. It is noteworthy that the metric underperforms on roles most traditionally male, e.g. plumber to electrician, perhaps due to the strong gender-occupation association overpowering our weaker gender-time association. Finally, we show similar plots for BERT and RoBERTa base and BERT large in Appendix D, but note the uncertainty metric appears more accurate for more overparameterized models.

## 6 EXTENDING TO MORE GENERAL SETTING

We now explore a more general problem space where the symbols in Figure 1 take on the following meanings: $G$ is the causal parent of $Y$, and $W$ is the non-causal parent of $Y$, yet nonetheless included because $W$ is a cause of both $X$ and $S$, where $S$ has the same meaning as before. We can thus partition any feature space into $G$, and candidates for $W$. A candidate can be validated as a suitable $W$ feature by checking for the conditional dependencies which we plot below. To make this hypothetical example slightly more concrete, we parameterize the DAGs in Figure 1, with the structural causal model (SCM) shown on the right.

Equation (5) and Equation (6) define $W$ and $G$ as independent exogenous 0-mean Gaussian noise, for which we set $\alpha = 10$ so that we can more easily trace the amplified noise through the DAG[9]. Equation (7) defines $S$ as an unweighted combination of $W$, $G$ and exogenous noise, with the selection mechanism setting all values above $2\alpha$ to 1, and to 0 otherwise, reducing the dataset to about 5% of its original size.

$$G := \alpha \mathcal{N}(0,1) \tag{5}$$

$$W := \frac{\alpha}{2} \mathcal{N}(0,1) \tag{6}$$

$$S := (W - G + \mathcal{N}(0,1)) > 2\alpha \tag{7}$$

$$X := W + \gamma G + \mathcal{N}(0,1) \tag{8}$$

$$Y := \gamma X + G + \mathcal{N}(0,1) \tag{9}$$

Equation (8) and Equation (9) we set $\gamma$ to 0 for the underspecified task and to 1 for the well-specified task, consistent with a 0 path weight for the grayed out arrows $G \to X$ and $X \to Y$ in Figure 1(b), and a full path weight for those same arrows in Figure 1(a).

Figure 6 plots the statistical relationships entailed by the SCM above. Columns (i) plots $X$ vs $Y$ for the unsampled, and columns (ii) plots $X$ vs $Y$ for the $S=1$ sampled distributions. In both Figure 6(a) and Figure 6(b), we can see that selection induced collider bias has little effect on the distributions (i) vs (ii) for the well-specified SCM, but causes the underspecified SCM's Pearson's $r$ coefficient to go from about 0 to 0.7. The latter is consistent with Section 4.2, in which entities such as gender and date (as well as gender and place) that are uncorrelated in the unsampled population, become spuriously correlated in LLMs trained on subsampled populations, where we claim the subsampling caused selection induced collider bias.

---

[9]We set different noise weights to $G$ and $W$ by arbitrarily dividing $\alpha$ by 2 in Equation (6), to reduce the likelihood of unintentionally constructing an unfaithful graph.

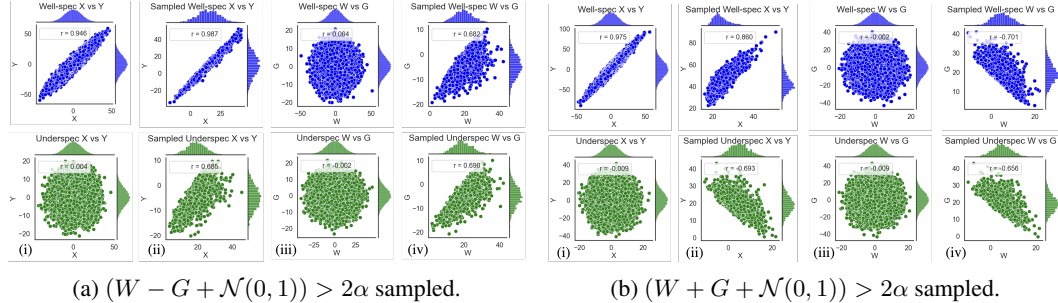

(a) $(W - G + \mathcal{N}(0,1)) > 2\alpha$ sampled.

(b) $(W + G + \mathcal{N}(0,1)) > 2\alpha$ sampled.

Figure 6: Statistical relationships induced by the SCM defined in Equation (5) to Equation (9), with Equation (7) separately defined above, with the well-specified (top row in blue) and underspecified (bottom row in green) DGP in Figure 1.

Columns (iii) and (iv) in Figure 6 plot the unsampled and the $S = 1$ sampled distributions for $W$ vs $G$. Comparing Figure 6(a) and Figure 6(b), we can see the direction of the correlation coefficient has been flipped for both the well and underspecified SCMs. However, the direction of the correlation coefficient flips in the $X$ vs $Y$ distributions of column (ii) only for the underspecified SCM. This is consistent with our derivation of $P(Y|X) \sim P(G|W)$ for underspecified tasks in Section 4.

In Figure 7, we replace Equation (8) with $X := \beta W + \gamma G + \mathcal{N}(0,1)$, where $\beta$ takes increasingly larger values: 1) $\beta = 0.01$ 2) $\beta = 0.1$ and 3) $\beta = 1$ (thus identical to Equation (8)), as a toy demonstration of $W$-injection. Similar to the uncertainty results in Section 5, we see underspecified tasks are more sensitive to $W$-injection, while well-specified are not.

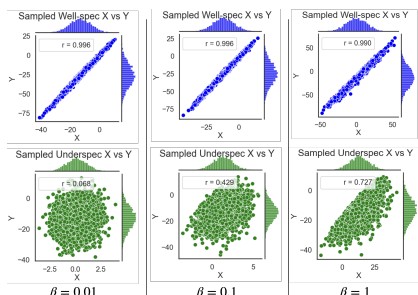

Figure 7: $W$-injection implemented via increasingly larger values of $\beta$ as the $W \to X$ path weight.

## 7 DISCUSSION AND REPRODUCIBILITY

As underscored in (D'Amour et al., 2020), the prevalence of underspecified tasks in machine learning requires the development of nuanced stress tests. In this work we strived to make our methods accessible and extensible to other tasks in machine learning. Please see more details in Appendix A.

We have argued that underspecified prediction tasks leave models vulnerable to selection induced collider bias which can result in the learning of otherwise non-interacting spurious associations, such as previously unreported associations between gender vs time and gender vs place, which we demonstrate on unmodified pre-trained BERT-like LLMs. We have introduced a lightweight inference-time technique for injecting spurious signals into prediction tasks to determine if the task is well-specified or underspecified, and demonstrated this in the form of an uncertainty metric on an established evaluation set. While not a universal solution, we hope our work can lead to the development of targeted heuristics, using our method to determine when the model is uncertain about a prediction task, and thus an alternate value should be returned (for example 'they' in the case of gender pronoun resolution).

For further research, we believe our work complements that of (Vig et al., 2020), which uses causal mediation analysis to intervene on LLMs at the individual attention head and neuron level to provide insights into the model's internal causal mechanisms mediating gender bias. Despite the more limited nature of our non-invasive investigation, our surprising empirical findings about the additivity of unrelated spurious associations is consistent with their surprising finding that gender bias appears decomposable between the elements of the direct and indirect effect within the model. We hope to be able to complement more of their internal model findings via external methods.

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

## A    REPRODUCIBILITY

Please see the following sources to reproduce the methods and measurements in this paper:

- Spurious Correlations Open Source Hugging Face Space (screen shot at Figure 10): https://huggingface.co/spaces/paper5186/spurious.
- Uncertainty Measurement Open Source Hugging Face Space (screen shot at Figure 11) : https://huggingface.co/spaces/paper5186/uncertainty.
- More General Setting Toy SCM : https://tinyurl.com/2ub4xyjs.
- Github repo to replicate all the plots in this paper: https://github.com/anon-anon-anony/sicb_paper.

## B    GENDER-IDENTIFYING WORDS

See Table 3 for the list of gender-identifying words that would contribute to total softmax probabilities masses accumulated for female and male genders for a given prediction in Figure 3. For example, if an LLM predicted 'her' in addition to 'she', we would sum their two softmax probabilities together for the final total softmax probability assigned to 'female'.

Table 3: A list of explicitly gendered words described in .

| MALE-VARIANT | FEMALE-VARIANT |
|---|---|
| HE | SHE |
| HIM | HER |
| HIS | HER |
| HIMSELF | HERSELF |
| MALE | FEMALE |
| MAN | WOMAN |
| MEN | WOMEN |
| HUSBAND | WIFE |
| FATHER | MOTHER |
| BOYFRIEND | GIRLFRIEND |
| BROTHER | SISTER |
| ACTOR | ACTRESS |

## C  $W$ VARIABLE X-AXIS VALUES

For {w} we require a list of values that are gender-neutral in the real world, yet due to selection induced collider bias are hypothesized to be a spectrum of gender-inequitable values in the dataset. For $W$ as *date*, we just use time itself, as over time women have become more likely to be recorded into historical documents reflected in Wikipedia, so we pick years ranging from 1801 - 2001. For $W$ as *place*, we use the bottom and top 10 World Economic Forum Global Gender Gap ranked countries (see details in Appendix C.1).

### C.1  PLACE VALUES

Ordered list of bottom 10 and top 10 World Economic Forum Global Gender Gap ranked countries used for the x-axis in Figure 3(c), that were taken directly without modification from `https://www3.weforum.org/docs/WEF_GGGR_2021.pdf`:

'Afghanistan', 'Yemen', 'Iraq', 'Pakistan', 'Syria', 'Democratic Republic of Congo', 'Iran', 'Mali', 'Chad', 'Saudi Arabia', 'Switzerland', 'Ireland', 'Lithuania', 'Rwanda', 'Namibia', 'Sweden', 'New Zealand', 'Norway', 'Finland', 'Iceland'

## D  EXTENDED WINOGENDER UNCERTAINTY RESULTS ON MORE LLMS

Figure 8 shows uncertainty results for all Winogender occupations where the masked pronoun is coreferent with the *professional*. Because the injected text (one of: {'man', 'woman', 'someone', 'other'}) is referring to the *participant* and not the *professional*, all these sentences are underspecified. The plots show that like RoBERTa large in Figure 5(a), RoBERTa base tends to report uncertainty results above 0 for most occupations, regardless of the word injected into the evaluation text for the *participant*, thus the model does not become erroneously certain about gender when the words 'man' and 'woman' are injected into the text. However, note that it is more difficult to see such a trend in BERT base and large.

Figure 9 shows uncertainty results for all Winogender occupations where the masked pronoun is coreferent with the *participant*, unlike Figure 8 where the pronoun is coreferent with the *professional*. Because the injected text (again one of: {'man', 'woman', 'someone', 'other'}) is referring to the *participant*, the sentences containing 'man' and 'woman' are well-specified, while the rest are underspecified. We see uncertainty results closer to 0 for most occupations when 'man' or 'woman' has been injected into the evaluation text for the *participant*, and generally above 0 otherwise, in particular for more highly overparameterized models like BERT large and RoBERTa base & large in Figure 5(b). It is more difficult to see this trend in BERT base.

## E  MODEL UNCERTAINTY DEMO

See Figure 11 for our open-source freely available demonstration where users can choose their own input text and select almost any BERT-like model hosted on Hugging Face to test for model uncertainty using selection induced collider bias induced spurious correlations.

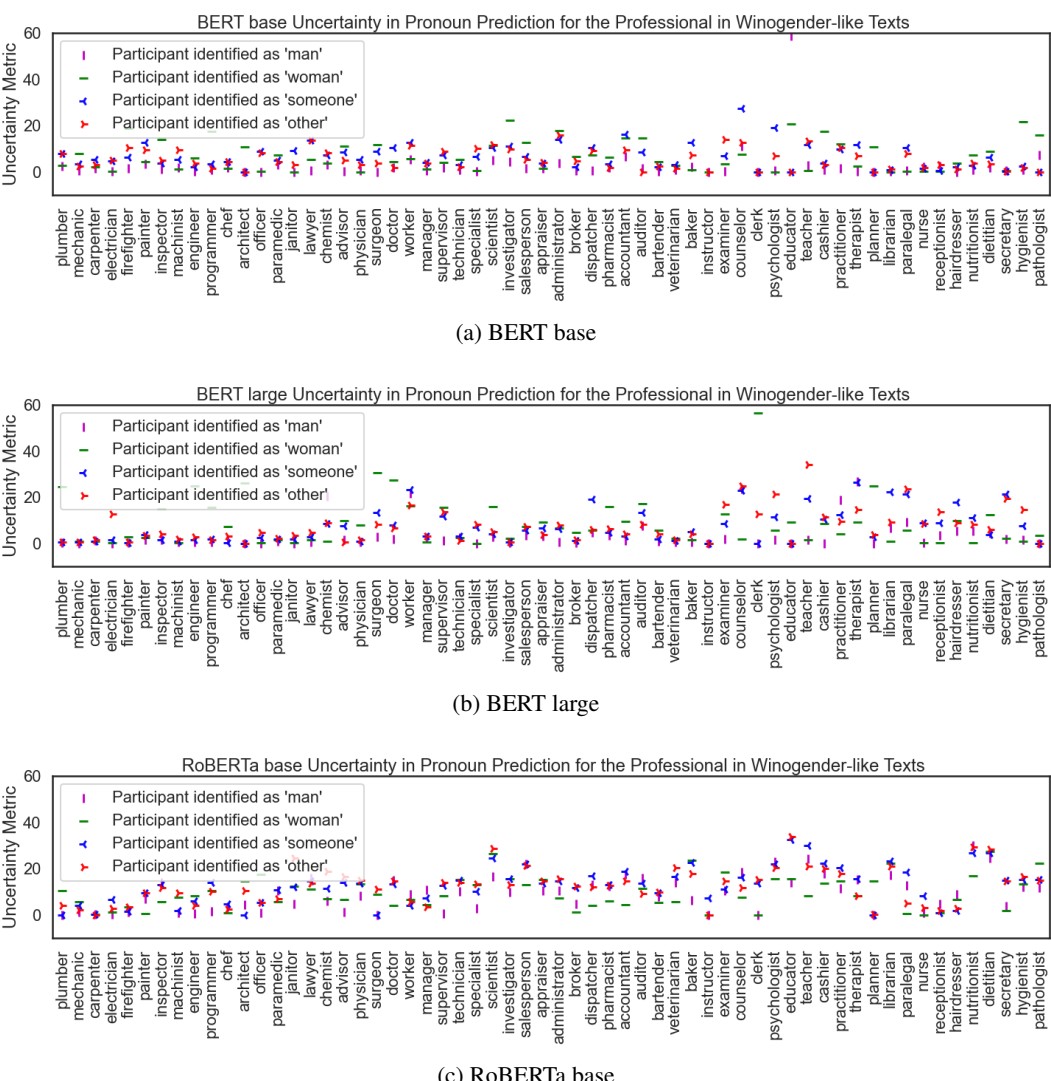

Figure 8: Uncertainty results for all Winogender occupations where the masked pronoun is coreferent with the gender-unidentified *professional*, thus all sentences are underspecified. These plots show that like RoBERTa large in Figure 5(a), RoBERTa base tends to report uncertainty results above 0 for most occupations, regardless of the word injected into the evaluation text for the *participant*, thus the model does not become erroneously certain about gender when the words 'man' and 'woman' are injected into the text. However, note that it is more difficult to see such a trend in BERT base and large.

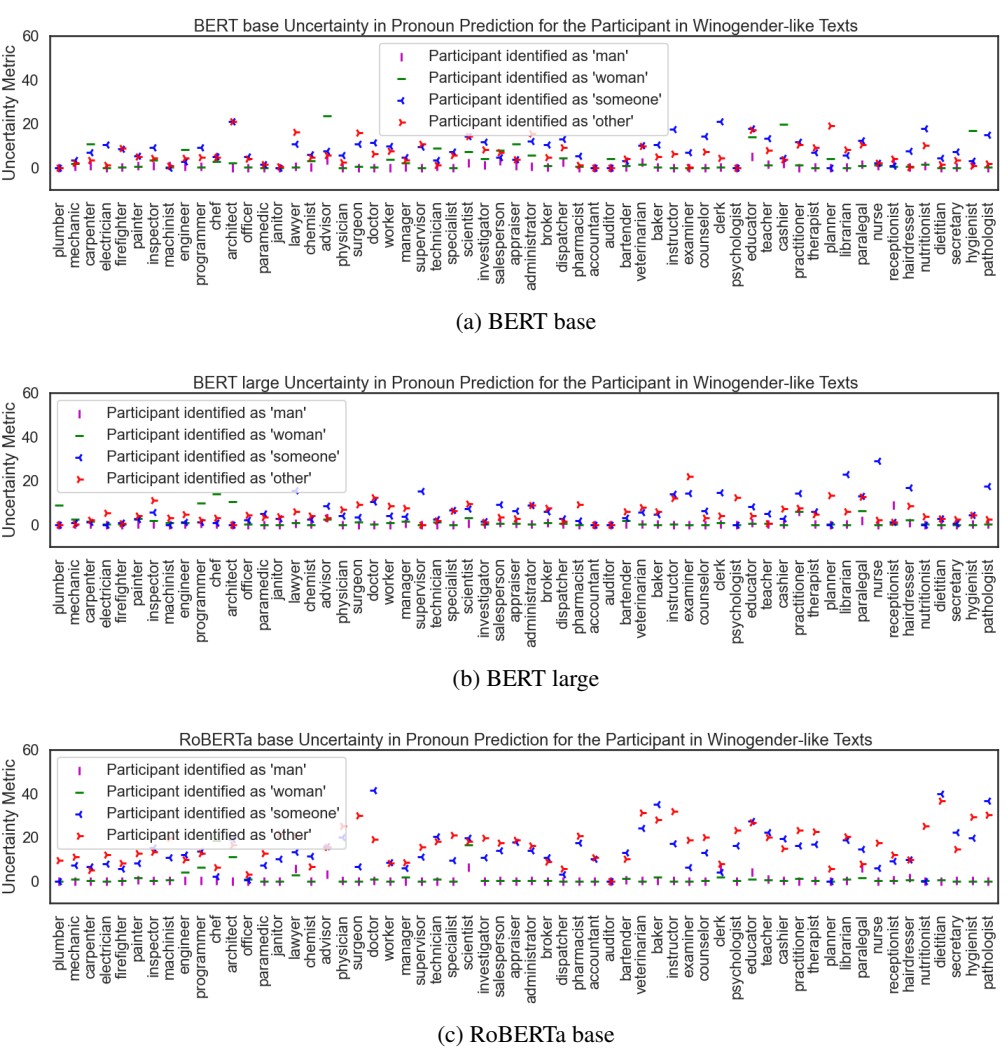

(a) BERT base

(b) BERT large

(c) RoBERTa base

Figure 9: Uncertainty results for all Winogender occupations where the masked pronoun is coreferent with the *participant*, thus the sentences containing 'man' and 'woman' are well-specified, while the rest are underspecified. We see uncertainty results closer to 0 for most occupations when 'man' or 'woman' has been injected into the evaluation text for the *participant*, and generally above 0 otherwise, in particular for more highly overparameterized models like BERT large and RoBERTa base & large in Figure 5(b). It is more difficult to see this trend in BERT base.

Click for date example inputs <-- x-axis sorted by older to more recent dates:

Click for country example inputs <-- x-axis sorted by bottom 10 and top 10 Global Gender Gap ranked countries:

Click for Subreddit example inputs <-- x-axis sorted in order of increasing self-identified female participation (see bburky):

Add-a-model example inputs <-- x-axis dates, with your own model loaded! (If first time, try another example, it can take a while to load new model.)

## Input fields

A) Pick a spectrum of comma separated values for text injection and x-axis.

A) Comma separated values for text injection and x-axis

GlobalOffensive, pcmasterrace, nfl, sports, The_Donald, leagueoflegends, Overwatch, gonewild, Futurology, space, technology, gaming, Jokes, dataisbeautiful, woahdude, askscience, wow, anime, BlackPeopleTwitter, politics, pokemon, worldnews, reddit.com, interestingasfuck, videos, nottheonion, television, science, atheism, movies, gifs, Music, trees, EarthPorn, GetMotivated, pokemongo, news, Fitness, Showerthoughts, OldSchoolCool, explainlikeimfive, todayilearned, gameofthrones, AdviceAnimals, DIY, WTF, IAmA, cringepics, tifu, mildlyinteresting, funny, pics, LifeProTips, creepy, personalfinance, food, AskReddit, books, aww, sex, relationships

B) Pick a pre-loaded BERT-family model of interest on the right.

Or C) select `add-a-model`, then add the mame of any other Hugging Face model that supports the fill-mask task on the right (note: this may take some time to load).

B) BERT-like model.

⚪ bert-base-uncased    ⚪ roberta-base    ⚪ bert-large-uncased    🔘 roberta-large

⚪ add-a-model

C) If you selected an 'add-a-model' model, put any Hugging Face pipeline model name (that supports the fill-mask task) here.

D) Pick if you want to the predictions normalied to these gendered terms only.

E) Also tell the demo what special token you will use in your input text, that you would like replaced with the spectrum of values you listed above.

And F) the degree of polynomial fit used for high-lighting potential spurious association.

D) Normalize model's predictions to only the gendered ones?
False

E) Special token place-holder
SUBREDDIT

F) Degree of polynomial fit
1

G) Finally, add input text that includes at least one gendered pronouns and one place-holder token specified above.

G) Input text with pronouns and place-holder token

She was a kid. SUBREDDIT.

## Outputs!

Hit submit to generate predictions!

Output text: Sample of text fed to model

<mask> was a kid.  WTF.

Probability of predicting female pronouns.

Probability of predicting male pronouns.

Figure 10: Demo where users can choose their own input text and select almost any BERT-like model hosted on Hugging Face to test for selection induced collider bias induced spurious correlations. https://huggingface.co/spaces/paper5186/spurious.

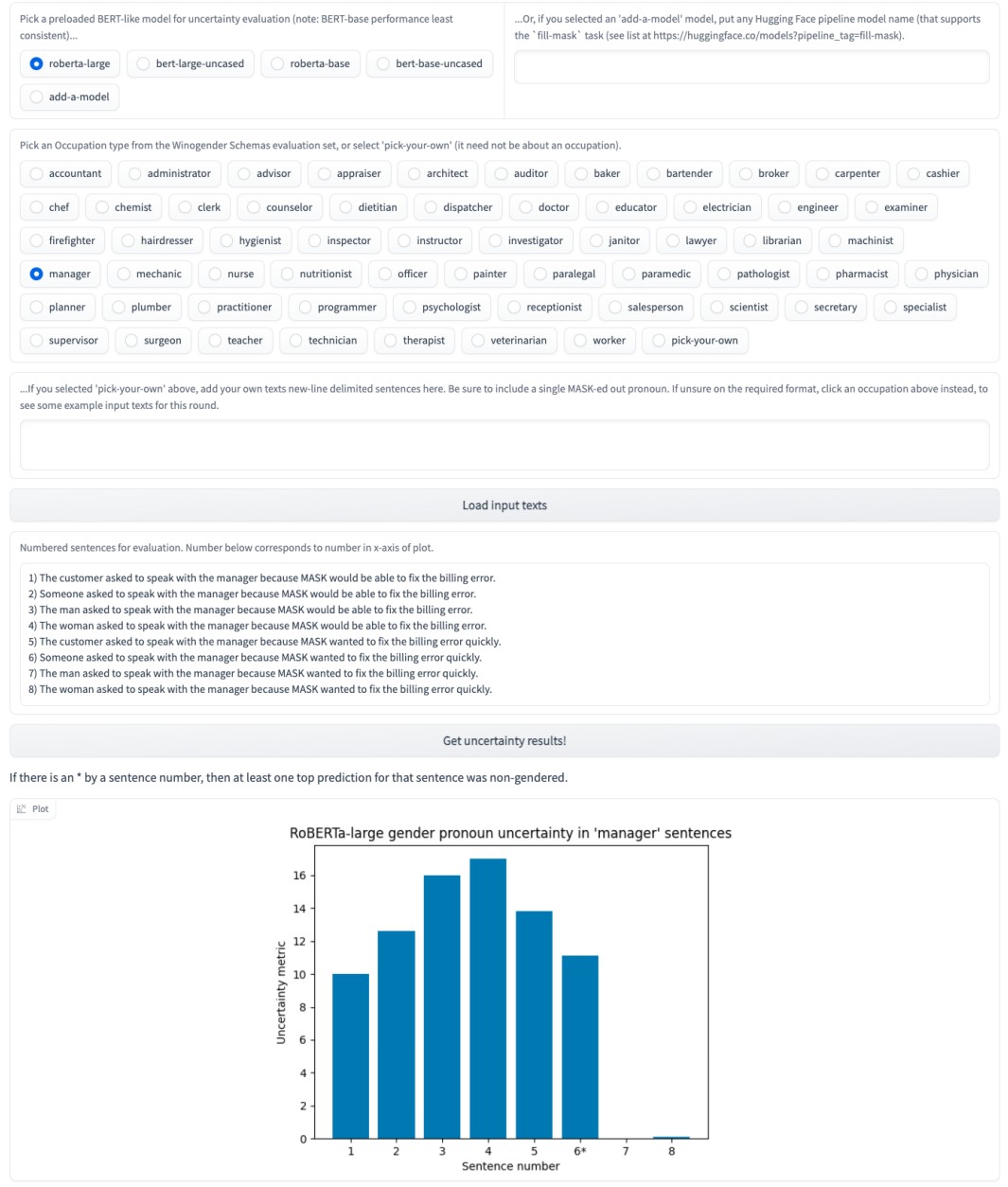

Figure 11: Demo where users can choose their own input text and select almost any BERT-like model hosted on Hugging Face to test for model uncertainty using selection induced collider bias induced spurious correlations. https://huggingface.co/spaces/paper5186/uncertainty.

