# OpenReview forum: "Selection Collider Bias in Large Language Models"
_ICLR.cc/2023/Conference — Submitted to ICLR 2023_

### Official Review · Reviewer_ukSe · 2022-10-19

**Confidence:** 3
**Correctness:** 3
**Technical Novelty And Significance:** 2
**Empirical Novelty And Significance:** 2
**Recommendation:** 5

**Clarity, Quality, Novelty And Reproducibility:**

Clarity: The paper is clearly written with minimal typos, the mathematical formulation is thoroughly motivated, and figures and equations are explained in detail.

Quality: The paper is of a very high quality. This includes thorough explanations, mathematically grounded problems formulations, use of external data sources, and training large models.

Novelty: It is unclear to this reviewer how novel of a contribution this work represents. The concepts of underspecification in machine learning models, and collider bias in graphical models, were already well-established concepts prior to this work. The novelty would be more obvious if this "selection collider bias" formulation led to a new correction or detection method, but that does not appear to be the case here. It is mentioned in the Discussion section that the correlations between gender and time/place are previously unreported, but this is used more as an example within the paper than a primary contribution.

Reproducibility: Reproducibility is helped by the authors open-sourcing their code and making available a nice UI for running similar experiments. However the paper fails to go into sufficient detail on the process they used to generate the data used for fine-tuning their models, or what the resulting data looked like, or what architectural changes were made to the models in order to train on this data. Even if the authors do not feel that this information is important enough to be present in the main body of the paper, it should at least appear in the appendix.

**Strength And Weaknesses:**

Strengths:

- The paper is well-organized and clearly written.

- The paper combines techniques from the usually-disjoint domains of causal modeling and LLMs.

- The paper takes pains to motivate the approach throughout its first several sections, both conceptually and mathematically.

Weaknesses:

- Referring to S as "access" is confusing, as this implies that it is an indicator of whether a given gender had access to the process that produced the dataset. However it's possible that a gender did have access, and yet the data was nonetheless highly gender biased, for example a corpus of harlequin novels written by female authors. It would therefore makes more sense to call this variable something like "representation", as what it actually expresses is the presence or absence of certain conditionally gendered data in the dataset.

- Need more information about how the dataset used to perform the fine-tuning in Figure 2 was constructed. Was there a heuristic used to identify sentences in an existing corpus that have a particular gender-ambiguous structure? Or was the heuristic used to construct the sentences directly? If the latter, then how were the "true" labels known? All of this information should be described in more detail in Section 3.1.

- The selection criteria of $(W ± G + N(0,1)) > 2\alpha$ used as a cutoff in the toy example of Section 8 seems like it would be throwing away a considerable majority of the W, G values. The authors should state the fraction of sampled values that satisfy the selection cutoff, preferably with an associated figure.

- The selection collider bias effect is presented as a general method, but (aside from the toy example) is presented solely in the context of gender biasing, to the point that one of the nodes is explicitly stated to be gender (G). The benefits of this approach would be clearer is it were demonstrated for other types of biases, for example bad weather being textually correlated with negative outcomes of all kinds, for causally unrelated reasons. For example "it was a [dark and stormy/sunny and bright] morning when the S&P 500 [crashed/rallied]".

- Part of the stated contribution of the paper is the introduction of a new technique for modeling uncertainty, however this measure is not investigated in much detail. Some natural additional steps would have been to (1) compute its correlation with a measure of intra-profession gender representation over time, (2) compare it to the standard deviation over softmax probabilities rather than taking their most extreme differences, (3) use it to identify specific scenarios in which a human observer might think that the LLM is conditioning on causally meaningful information, but really is conditioning on acausal correlations.

- Although the paper centers around a novel data biasing mechanism, it does not make any attempt to describe how one might go about alleviating this biasing, or detecting it a priori. (The proposed uncertainly measure is only useful if you already know where to look.)

Typos:

- Bottom of page 3: "we could the use"

- Top of page 5: "supports our supports our"

- Top of page 9: "only holds underspecified models"

- Middle of page 11: opening-quote marks in list of countries are backwards

- Figure 3: Pearson's r should be lower-case in the legend

**Summary Of The Paper:**

The paper "Selection Collider Bias in Large Language Models" explores a type of collider bias in which the shared effect variable is the selection process by which training data is generated. They show that the effect of this selection process can be highly pronounced in cases of textual ambiguity, leading to variables that are causally disconnected in the real word (e.g. gender vs year) exhibiting strong predicted correlations. They use this spurious correlation to define a novel uncertainty metric that can help indicate when a model is conditioning on non-causal variables. They also demonstrate the validity of their model formulation on a more general toy example. The authors also open-sourced their code with an easy-to-use UI so that others can easily experiment with it.

**Summary Of The Review:**

Overall the paper is clear, well-written, and well-organized. However the primary contribution of the paper is a novel mathematical formulation of the already well-established phenomenon that if a model cannot predict a label by conditioning on causally-related words, then the model will try to predict the label using whatever non-causal word correlations are present, and these may be sampling artifacts. The authors' new proposed formalism for describing this phenomenon does not lead to a new method for diagnosing it a priori, nor does it lead to a new approach to alleviating it. Therefore this reviewer does not feel that this paper meets the bar for acceptance.

---

### Official Review · Reviewer_qdVm · 2022-10-22

**Confidence:** 3
**Correctness:** 2
**Technical Novelty And Significance:** 2
**Empirical Novelty And Significance:** 2
**Recommendation:** 3

**Clarity, Quality, Novelty And Reproducibility:**

Aspects of the paper (definition of concepts in the introduction, for instance) are pretty clear, while others (relationship to existing work, some key methodological details, motivations, intended contributions, etc) are not. I don't think the methods are clear enough for sufficient reproducibility, and the lack of a Related Work discussion makes novelty also unclear. Overall quality is diminished by all of the issues I've discussed, and can be improved by addressing them.

**Strength And Weaknesses:**

Strengths: The paper involves the important topic of spurious correlations, and it frames that topic within an interesting causal inference framework. The paper identifies spurious correlations between model gender preference and gender-irrelevant cues like date and country.

Weaknesses: The paper makes no effort to contextualize its contribution with respect to related work. The paper cites a bit of work while explaining and motivating the relevant concepts and formalisms that it uses, but there is no Related Work section to lay out how the paper's contribution relates to and is novel with respect to all of the other literature in this area.

It is not only the contextualization of the contribution that the paper doesn't make clear -- the paper doesn't really make a clear argument for what contribution it is making in the first place. The first symptom of this is that the introduction spends a lot of time defining "selection collider bias" and briefly arguing for its relevance, but makes no mention at all of what contribution the paper plans to make with respect to this selection collider bias concept. The paper then proceeds to present a number of theoretical discussions and empirical demonstrations, so in theory there could be good contributions that should simply be laid out more explicitly (in the introduction and elsewhere) -- but at the moment I don't think the paper is making any clear statement for what exactly it's trying to accomplish, and as I elaborate in the next paragraph, this contribution isn't clear from the content of the experiments/theoretical discussions either.

Examining the experiments themselves to extract a potential contribution, it seems to me that the most concrete finding is that pre-trained LMs' preference for different pronoun genders is influenced by irrelevant cues like year and country mentions, but more so/exclusively so when actual relevant gender cues are not available/sufficient to inform the prediction. This characterization seems reasonably clear from the second experiment -- but the first experiment is harder to interpret, in large part because the specific details of the data and task used for fine-tuning are not actually provided, so it is difficult to know exactly what to make of the stronger correlations for the fine-tuned models. But the fact that the original LMs also show some spurious correction between year/country and gender preference is somewhat interesting. However, there are numerous issues surrounding this potentially interesting finding: 1) it's not clear whether it's a novel finding, since there is no comparison to related literature, 2) the paper does not provide any particular motivation for having chosen date and country as possible spurious correlating variables with gender, apart from speculating that they could show spurious correlations (wrt this, see also below), and 3) the paper puts no particular focus on this finding and makes no argument about its significance.

It seems likely that the authors want to focus more on a theoretical contribution that emerges from conceptualizing spurious correlations within the causal framework that they use. While I'm happy to be convinced otherwise on this, I'm also just not seeing any clear, novel insight that emerges from the theoretical framing or reasoning here -- mainly it seems just to be restating and reframing the existence of spurious correlations. Additionally, some of the theoretical framing strikes me as unclear and potentially problematic: the specific example I have in mind is the conceptualization of variable G in Section 3, which is described as corresponding to "gender", but it is not clear whether this refers to gender of the writer of the text or gender of the entity being referred to by a pronoun or pronouns in the text. Assumption of a causal relationship between G and Y seems to suggest G refers to gender of entities referred to by pronouns, but the discussion of a relationship between G and dataset access seems to suggest G refers to gender of writers of text. Obviously these two variables cannot be conflated, so the authors should clarify their thinking here.

**Summary Of The Paper:**

This paper focuses on what it refers to as "selection collider bias", using a causal inference framework to discuss spurious correlations in language data. The paper seems largely focused on formalizing notions of selection bias and spurious correlations through the language of causal inference, and presenting reasoning about factors involved in these phenomena. Concretely the paper offers a couple of experiments: the first involves fine-tuning LMs by "masking common gender-identifying words for prediction", and showing that both these fine-tuned models and normal pre-trained LMs show an interaction between gender pronoun preference and mentions of years or of countries. The second experiment involves augmenting the WinoGender dataset with prefixes mentioning a date, as well as explicit mentions of gender for one of the antecedent entities, and showing that there are correlations between the date and gender preferences for the pronoun, but only when the pronoun's antecedent is not explicitly specified. The third experiment runs a toy simulation to further demonstrate the relationship between data, label, and causal variables depending on underspecification and sampling.

**Summary Of The Review:**

The paper focuses on an important overall problem, uses an interesting theoretical framing, and has at least one seemingly interesting empirical finding. However, it does not attempt to contextualize with respect to related literature, does not provide a clear statement (or demonstration) of what contribution it is making, and has additional issues with methodological and theoretical clarity.

---

### Official Review · Reviewer_LK4n · 2022-10-24

**Confidence:** 3
**Correctness:** 3
**Technical Novelty And Significance:** 3
**Empirical Novelty And Significance:** 3
**Recommendation:** 5

**Clarity, Quality, Novelty And Reproducibility:**


[1] Definition of sample selection bias and selection collider bias.

Lines 5-6 in Intro. say "sample selection bias occurs when some mechanism, observed or not, causes samples to be included or excluded from the dataset".  Please give more explanation about sample selection bias.  From my view, it seems that "samples to be included or excluded" defines a full set where the probability is always 100%.  Given any mechanism, this event always happens.

Figure 1 illustrates what is selection collider bias. Is there any example to explain selection collider bias?

[2] Definition of underspecification

Section 2 describes what is underspecification. Compared with robustness and over-fitting, underspecification is somehow not very common. It would be better to just give an illustrative example or illustrative definition.

[3] Details of Figure 2

About "Now applying the above-mentioned Markov and faithfulness assumptions to the underspecified model in Figure 1(b), we can estimate the conditional probability of a gender-identifying word, Y, given gender-neutral text, X, in a LLM as shown in Equation (1) through Equation (4)", could the author provide more details?

[4] What is " the lowest relative uncertainty" in Table 2?

[5] Details of uncertainty definition in Section 7.2




**Strength And Weaknesses:**

Strength:

It is interesting to see how LMs learn spurious bias from causal mechanisms.





Weaknesses:

[1]  There are important concepts lacking illustrative explanations. It makes the whole paper hard to read.


**Summary Of The Paper:**

This paper explores the spurious correlation problem, a common problem in current LMs that learns dependency relations between two unconditional entities. The authors take "date", "place", and "gender" as examples and analyze the spurious correlation between gender and date (or place). The authors also provide an uncertainty metric to evaluate and identify spurious correlations.

**Summary Of The Review:**

The paper focuses on an interesting problem that explores spurious correlation from a causal mechanism perspective. However, due to missing necessary explanations, it is hard for me to understand the details of the paper.

---

### Author Response · Authors · 2022-11-16
**Issues with Hugging Face’s Spaces hosting of our demos**

We just became aware of some issues with Hugging Face’s Spaces hosting of our demos.

In our several months of using Spaces for these demos, we have never run into any issues, so this is surprising and unfortunate timing.

We are reaching out to the Hugging Face Gradio team to work towards a speedy resolution and looking into alternative hosting options. We will post updates as we have them.

Our sincere apologies for the inconvenience.

---

> ### Author Response · Authors · 2022-11-16
> **Issues resolved. Demos working.**
>
> The issue appears to have been a dramatic increase in rendering time (about a 100x increase), which may have been only temporary.
>
> We have nonetheless upgraded the hosting hardware during this crucial time and linked to a backup colab notebook in the spurious correlations demo.
>
> Our apologies again for any inconvenience this may have caused.

---

### Author Response · Authors · 2022-12-05
**To all reviewers and AC, during Discussion Stage 2**

Dear Reviewers and AC, We realize this is likely an incredibly busy time for all, however, we wanted to check in during this stage of the discussion process, with the hope that we may be able to offer any remaining or further clarifying statements regarding our work.

We were very pleased to read that the reviewers generally found our paper to be addressing an important topic from an interesting causal perspective.

We were also very appreciative of the critical feedback regarding the confusing layout of our paper, which lacked clear statements regarding the contributions of our work.

Our subsequent revision of the paper included a bulleted ‘Contributions’ section (that had previously been entirely lacking), as well as “Related Work” subsections to help contextualize our contributions.

As reviewer qdVm fairly stated regarding the original revision: “While I'm happy to be convinced otherwise on this, I'm also just not seeing any clear, novel insight that emerges from the theoretical framing or reasoning here -- mainly it seems just to be restating and reframing the existence of spurious correlations.”, we hope our final version makes clear our novel insights, and is convincing in its distinction from existing literature on spurious correlations.

---

### Decision · Program_Chairs · 2023-01-20

**Decision:**

Reject

**Justification For Why Not Higher Score:**

Unfortunately, reviewers were not convinced the technical novelty was there. The topic is also somewhat marginal.

**Justification For Why Not Lower Score:**

Cannot be any lower.

**Metareview: Summary, Strengths And Weaknesses:**

This paper has attracted a but of discussion among the reviewers and the authors. Overall the reviewers engaged
with the authors and provided excellent feedback which the authors acknowledged. The problem still remains that the
paper in its original form lacked clarity and assumed familiarity with prior art on collider selection bias. In my view the
paper should be rewritten and be made more accessibly. Moreover, reviewers were not convinced about the novelty
of the contribution.

**Summary Of Ac-Reviewer Meeting:**

This was not a borderline paper.